

# Effects of root phenotypic changes on the deep rooting of *Populus euphratica* seedlings under drought stresses

Zi-qi Ye[1,2], Jian-ming Wang[1], Wen-juan Wang[1], Tian-han Zhang[1] and Jing-wen Li[1]

[1] The College of Forestry, Beijing Forestry University, Beijing, China
[2] Institute of Forest Ecology, Environment and Protection, Chinese Academy of Forestry, Beijing, Beijing, China

## ABSTRACT

**Background:** Deep roots are critical for the survival of *Populus euphratica* seedlings on the floodplains of arid regions where they easily suffer drought stress. Drought typically suppresses root growth, but *P. euphratica* seedlings can adjust phenotypically in terms of root-shoot allocation and root architecture and morphology, thus promoting deep rooting. However, the root phenotypic changes undertaken by *P. euphratica* seedlings as a deep rooting strategy under drought conditions remain unknown.

**Methods:** We quantified deep rooting capacity by the relative root depth (RRD), which represents the ratio of taproot length to plant biomass and is controlled by root mass fraction (RMF), taproot mass fraction (TRMF), and specific taproot length (STRL). We recorded phenotypic changes in one-year-old *P. euphratica* seedlings under control, moderate and severe drought stress treatments and assessed the effects of RMF, TRMF, and STRL on RRD.

**Results:** Drought significantly decreased absolute root depth but substantially increased RRD via exerting positive effects on TRMF, RMF, and STRL. Under moderate drought, TRMF contributed 55%, RMF 27%, and STRL 18% to RRD variation. Under severe drought, the contribution of RMF to RRD variation increased to 37%, which was similar to the 41% for TRMF. The contribution of STRL slightly increased to 22%.

**Conclusion:** These results suggest that the adjustments in root architecture and root-shoot allocation were predominantly responsible for deep rooting in *P. euphratica* seedlings under drought conditions, while morphological changes played a minor role. Moreover, *P. euphratica* seedlings rely mostly on adjusting their root architecture to maintain root depth under moderate drought conditions, whereas root-shoot allocation responds more strongly under severe drought conditions, to the point where it plays a role as important as root architecture does on deep rooting.

Corresponding author
Jing-wen Li,
lijingwenhy@bjfu.edu.cn

## INTRODUCTION

The root system is the main plant organ acquiring below-ground resources. To adapt to the inherent heterogeneity of soil resources, plants can adjust root phenotype on different integrated levels (*Chapin, 1991*; *Nicotra et al., 2010*; *Poorter et al., 2012*): they can change their relative investment of biomass in shoots and roots on an individual-level, they can adjust root system architecture on an organ-level, or alter root morphology on a module-level (Fig. 1). Most likely, plants adjust their roots on all three levels. However, different environmental stresses may result in different degrees of root phenotypic adjustments, resulting in different magnitudes of functional contributions for plants under environmental stress. For instance, *Freschet, Swart & Cornelissen (2015)* found that root mass fraction (RMF; the proportion of total plant mass allocated to roots) responded more strongly than specific root length (SRL; root length for a given unit of plant mass) to nutrient deficiency, thus suggesting that with nutrient stress, the increased allocation to roots seems more important than the root morphological change for the plant to achieve an increase in root length. Nevertheless, root architectural change is a basic way to improve fine-root function efficacy (*Lynch, 1995*; *McCormack et al., 2017*). Increasing the density and length of the distal roots can increase root absorption area (*Kong et al., 2014*). Additionally, different root patterns can influence plant uptake efficiency in heterogeneous resource environments (*Lynch, 2005*) and can affect the ability of root systems to capture relatively immobile vs mobile soil resources (*Fitter, 1987*). However, few studies have focused on the relative functional importance of the different phenotypic changes in stressed plants, especially the relative role of root architectural changes (*Weemstra et al., 2016*; *Kramer-walter & Laughlin, 2017*; *Freschet et al., 2018*).

*Populus euphratica* Oliv. (Salicaceae) is a dioecious riparian tree species found discontinuously within the continental-arid climate region of Central Asia (*Browicz, 1977*; *Wang, 1996*), which forms monospecific stands along continental rivers. This poplar is an obligate phreatophyte with a root system that continuously contacts the groundwater or the soil water-saturated zone (*Zhu et al., 2009*), meaning that its growth and survival depends highly on locating and acquiring groundwater (*Gries et al., 2003*). However, its seedlings cannot reach groundwater during the early stages, meaning that under the harsh environments in arid regions, *P. euphratica* can only propagate generatively in the freshly deposited floodplain soils from May to August (*Cao et al., 2012*). Even so, the optimal soil conditions in such floodplains for seedling germination and growth only exist for a short time, as the surface soil post-flood rapidly becomes increasingly dry and salty due to evaporation. The rapid onset of winter also shortens the growth period for the seedlings. As a result, *P. euphratica* seedlings must establish a deep root system during the early stage, as only seedlings with deep roots can secure water uptake during the dry period and survive the following year (*Zerbe & Thevs, 2011*). According to *Thevs et al. (2008)*, *Wiehle et al. (2009)*, and our previous field investigations, *P. euphratica* seedlings established in floodplains possess deep root systems asymmetrical in size to the shoots.

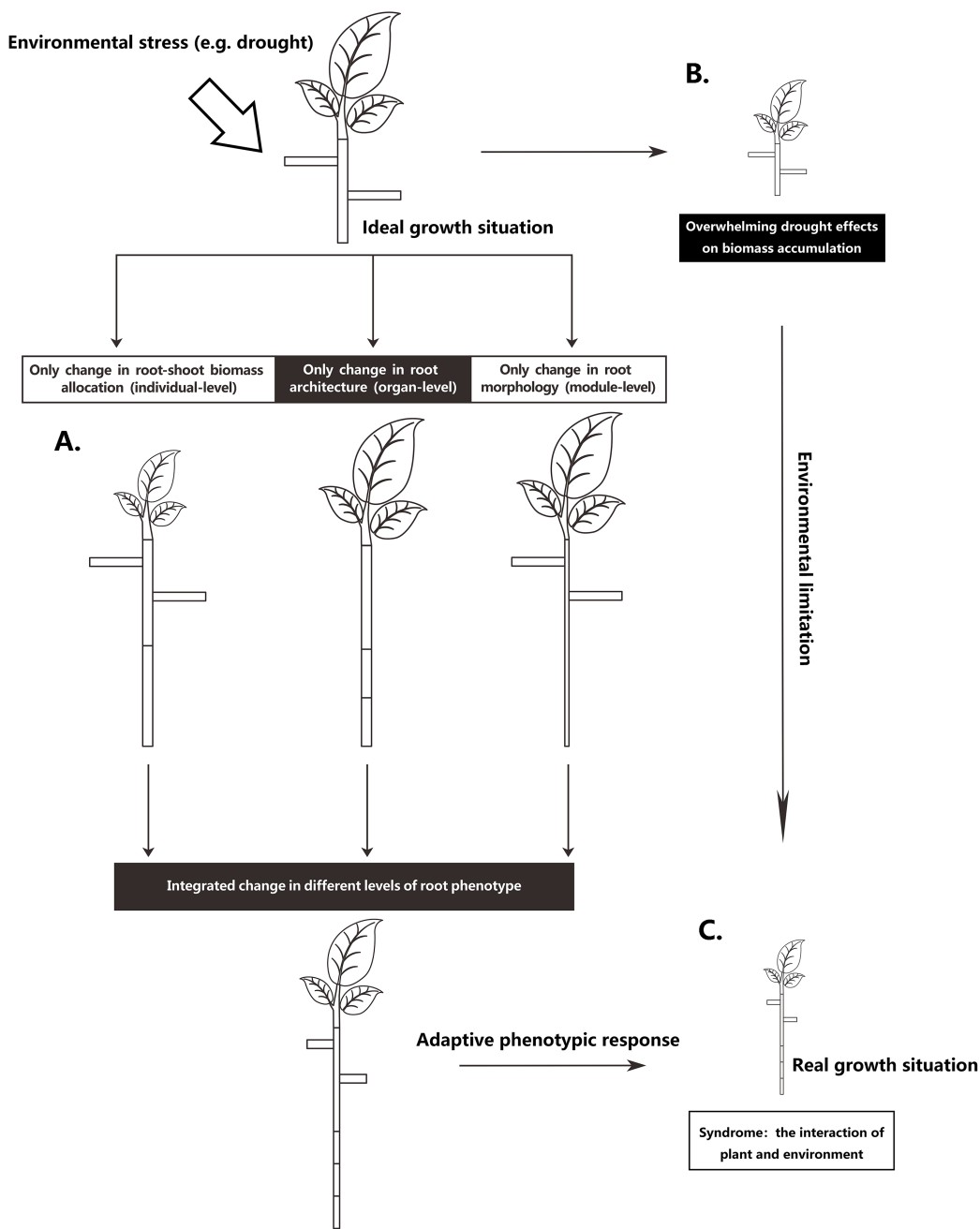

**Figure 1 Schematic representation of partitioning the total drought effect into environmental limitation of growth and different levels of root phenotypic responses.** (A) A scheme to increasing a certain root length through three levels of root phenotypic changes, respectively. To achieve an increase of certain root length, plant can phenotypically adjust at three levels, respectively, but most commonly, plant roots adjust at all three levels when they respond to drought. (B) Drought exerts considerable influence in metabolism, causing an overall limitation of growth. (C) Real growth situation is a syndrome from environmental limitation and multiple facets of integrated phenotypic responses.

Therefore, deep rooting is considered a key process for survival and establishment of early *P. euphratica* seedlings in an inconsistent riparian environment (*Hukin et al., 2005*; *Wang et al., 2015*).

*Populus euphratica* seedlings are easily subjected to drought stress in semi-arid regions (*Thevs et al., 2008*; *Stella et al., 2010*), which decreases biomass accumulation and limits root development into the deep soil. Accordingly, they take some phenotypic adjustments in order to deal with drought stress, such as increased root biomass allocation (*Wang et al., 2015*), root architecture adjustment and root morphology regulation (i.e., increased SRL; *Lü et al., 2015*). Each of these changes was believed to cause increased seedling root depth. However, as previously outlined, different levels of phenotypic changes may have different functional contributions, and thus our knowledge is still incipient regarding the exact effects of root phenotypic changes on the root depth of *P. euphratica* seedlings under drought stress. Further research on this issue would aid in our understanding of the drought adaptive strategy of *P. euphratica* and other similar deep-rooted plants in the arid regions.

To understand the changes to plant root depth in response to environmental conditions, it is important to distinguishing between absolute and relative plant dimensions because water deficit may strongly influences the overall plant size in a negative manner (Fig. 1), which often obscures the underlying deep-rooting process (*Schenk & Jackson, 2002*). Therefore, we adopted the absolute root depth to total plant biomass ratio to quantify relative root depth (RRD), that is, deep-rooting capacity, which follows the method of *Ryser & Lambers (1995)* who used root length per total plant biomass to express the relative length of fine roots, that is, root uptake capacity. Here, we define deep-rooting capacity as the capacity of the plant use all available resources to achieve an increase in root depth via all possible ways. Deep-rooting capacity is regarded as an overall capacity that could be depicted by several facets of drought responses (this definition is similar to that of root nutrient acquisition capacity used by *Freschet et al. (2018)*). Thus, RRD mathematically follows from the increases in RMF and/or the proportion of root system mass invested in taproot (i.e., taproot mass fraction (TRMF)) and/or the taproot length achieved per unit taproot mass (i.e., specific taproot length (STRL)) (see Eq. 1).

The present study aimed to answer the following questions: (1) How do *P. euphratica* seedling roots change phenotypically on the individual-, organ- and modular- levels under drought stress? (2) Which level of root phenotypic adjustment (root-shoot allocation, root architecture, or morphology) plays a leading role in facilitating the deep rooting of the drought-stressed seedlings? (3) Does the relative contributions of these different root phenotypic adjustments on deep rooting vary with drought intensity?

# MATERIALS AND METHODS
## Experimental method
### Nursery phase
This study was conducted at the State Forest Farm located in Ejin Banner, Inner Mongolia, China. All seeds were randomly collected from a mature natural forest dominated by *P. euphratica* (41°57 51.3″N, 101°05 06.0″E) along the Ejin River. Planted pots were 20 L in volume, 40 cm in depth, and filled with 16 kg (dry weight) of substrate—a 4/6 (v/v) mixture of peat and sand. A slow-release fertilizer (four g L$^{-1}$ Osmocote 16:9:12 NPK and

trace elements, product code: 8840) was pre-mixed within the substrate. From May to June 2016, *P. euphratica* seeds were sown and germinated in 90 plastic pots placed in a greenhouse. After a month of growth with normal management, three seedlings per pot, averaging 2.0 ± 0.5 cm in height and bearing four to six leaves, were selected and transferred to the open-air nursery. After 2 weeks of acclimation, only a single healthy seedling per pot, averaging 7.0 ± 1.5 cm in height and 0.9 ± 0.2 mm in ground diameter, was kept. This nursery phase took about 50 days.

### Experimental treatments

This experiment was conducted over a period of 60 days from July to September 2016, during which the average of daily minimum and maximum temperatures were 21.7 and 34.3 °C, respectively. The daily maximum temperature ranged from 25 to 44 °C. The experiment was conducted in a completely randomized design, including three water treatments: 70–80% of field capacity ((optimal water content (OW)), 50–60% of field capacity (moderate drought stress (MD)), and 30–40% of field capacity ((severe drought stress (SD)). Each treatment was replicated 30 times, and each replication consisted of a seedling planted in an independent pot.

The seedlings were randomly assigned to one of the three treatments to avoid the variance in individual growth. Pot body was completely hidden into ground to avoid pot warming and just exposed surface soil to the arid climate, by which the soil in pot could have the relative real underground temperature and moisture in vertical gradient. Because the evapotranspiration rate in Ejin is high during summer days, soil water content was supplemented every day using the weight method to keep it at a certain range. After water supplementation, the pots were rearranged randomly to neutralize the influence of potential environmental heterogeneity. Plastic film was used to cover the nursery to avoid rainfall. Owing to the death of five seedlings subject to the SD treatment, and to the destruction of two seedlings from OW treatment and two seedlings from MD treatment during destructive sampling and root measurements, only 81 seedlings were successfully measured and accessed to analyses.

### Growth measurements and destructive sampling

Intact root systems were cleaned of soil with a gentle water jet while a sieve was used to collect any root fragments detached from the system during this process. The cleaned root systems were then floated on water in a transparent tray and imaged using an Epson Expression Perfection V850 Pro Scanner with 800 DPI resolution (Fig. 2). Adobe Photoshop CS6 software was used to reduce image noise and black margins. The image analysis software (Win-RhIZO 2013a; Pro Instruments Inc., Québec, Canada) was used to analyze images and to estimate the length of total, lateral, and distal roots, the average diameter of the total roots, the number of lateral roots, as well as the external path length ($p_e$), magnitude ($\mu$), and altitude ($a$) (Table 1). The taproot was then detached from the root system with scissors, and its length and dry mass were determined. Lateral roots were defined as the root segments connected to the taproot with a root order >3, so as to guarantee that such roots perform conducting and foraging functions.

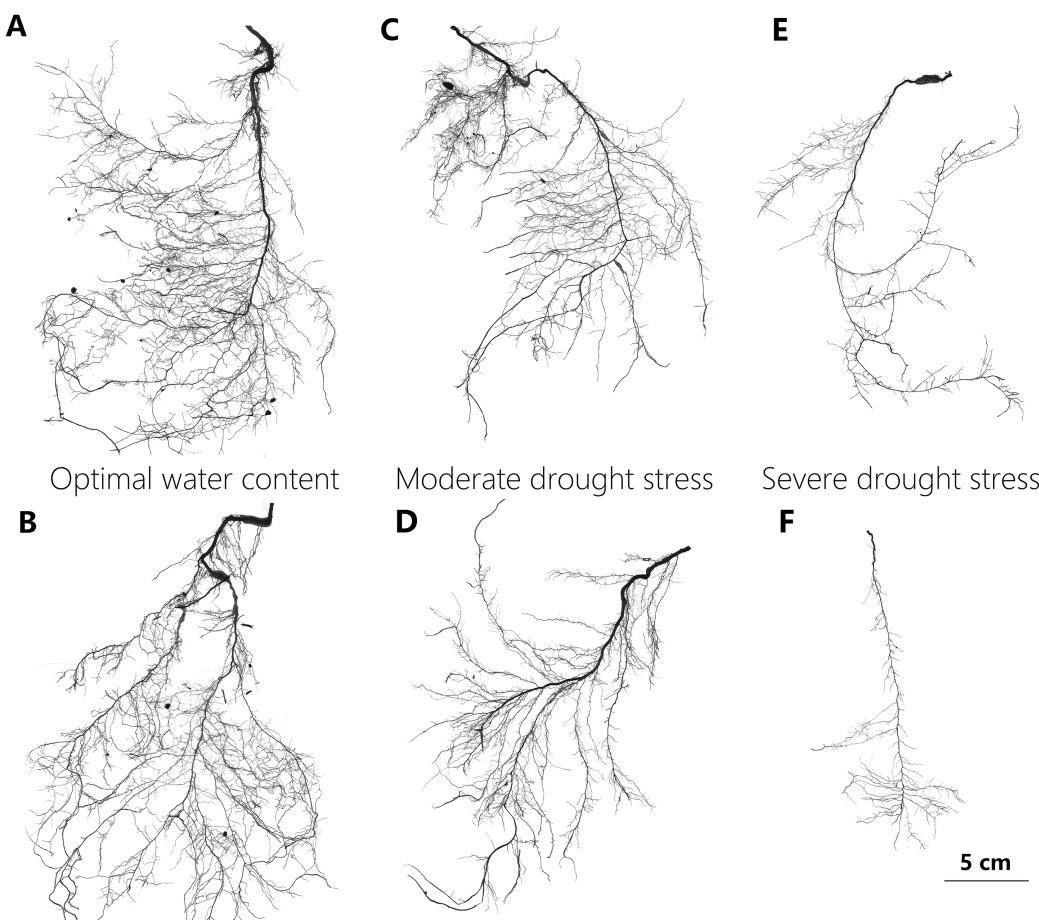

A

Optimal water content

B

C

Moderate drought stress

D

E

Severe drought stress

F

5 cm

**Figure 2 Representative scan images used in the study to illustrate changes in root system of *P. euphratica* seedlings with drought treatments.** The representative root images of *P. euphratica* seedlings show the relatively denser branch, deeper root depth and more herringbone-like branching pattern of root with drought increasing. (A and B) Show the root images from the seedlings under OW, in which their TI = 0.697, DBI = 0.056, and TI = 0.650, DBI = 0.032, respectively. (C and D) From the seedlings under MD, in which their TI = 0.702, DBI = 0.074, and TI = 0.702, DBI = 0.086, respectively. (E and F) From the seedlings under SD in which their TI = 0.754, DBI = 0.185, and TI = 0.773, DBI = 0.258, respectively. OW, optimal water content; MD, moderate drought stress; SD, severe drought stress. DBI is the dichotomous branching index and TI is the topology index, and both of them are used to quantify root architectural changes (for detailed definition see table 1). A DBI of zero is characteristic of a perfectly dichotomous branching structure and similarly, a TI close to 0.5 means a dichotomous-like branching structure. Contrastly, both a DBI of one and a TI of one represents a perfectly herringbone branching structure.

Three intact lateral root branch segments with over three levels of root order were selected randomly and scanned to estimate the average length and diameter of the distal roots. Using a scalpel, all distal fine roots were dissected and was determined with its dry mass. If it was impossible to meet these criteria, the whole root system was measured. The dry weights of the taproot, sampled distal roots, and the other root parts were obtained by air drying plant matter in an oven at 72 °C for 60 h. Total root biomass, SRL, biomass-related variables, and two topological indices were calculated using these dry weight values (Table 1).

**Table 1 Calculated root variables and its abbreviations, units, description and functional role.**

| Root variables (Abbrev.) units | Description | Functional role |
|---|---|---|
| Relative root depth (RRD) cm taproot (g plant)$^{-1}$ | Taproot length per total plant dry mass | Capacity for deep rooting |
| Root mass fraction (RMF) g root (g plant)$^{-1}$ | The proportion of total plant mass allocated to roots | Total plant resources allocated to root functions (e.g., absorption, transport, foraging, anchoring) *Markesteijn & Poorter (2009)* |
| Taproot mass fraction (TRMF) g taproot (g root)$^{-1}$ | The proportion of total root mass allocated to taproot | Total root resources allocated to the taproot functions (e.g., anchoring, foraging, transport) |
| Specific taproot length (STRL) cm taproot (g taproot)$^{-1}$ | Taproot length per amount of taproot biomass invested | The efficiency of taproot resources used to deep rooting |
| Lateral root branching density (LRBD) number lateral root (cm taproot)$^{-1}$ | Lateral roots branching number per unit of taproot length | Potential capacity of exploration and exploitation to horizontal soil resources |
| Topology Index (TI) | $TI = \log_{10}(a)/\log_{10}(\mu)$; altitude $a$ is number of links in the longest path from base to tips; magnitude $\mu$ is number of external links or the number of root tips *Glimskär (2000)* | TI vary from one close to 0.5, and DBI vary between zero and one, both with large values indicative of a more herringbone-like root system that are thought to be more efficient at intercepting mobile resources, such as water, by extensive soil exploration, contrasted with dichotomous-like systems that are better at acquiring immobile resources, such as phosphorus by intensive soil exploration *Fitter (1987)* |
| Dichotomous branching index (DBI) | $DBI = (p_e - \min(p_e))/(\max(p_e) - \min(p_e))$; $p_e$ is the sum of the number of links in all paths from each external link to the base link; $\max(p_e)$ and $\min(p_e)$, respectively is the theoretical external path length for a system of given magnitude that has a completely herringbone and dichotomous topology, for detailed calculation see *Šmilauerová & Šmilauer (2002)* | |

## Root phenotypic adjustments and deep-rooting capacity analysis

As in many other studies, root-shoot allocation was quantified as the fraction of plant biomass invested in the roots (RMF; see Table 1 for definition), and taproot morphological change could be expressed as STRL (see Table 1 for definition). These two aspects of root phenotypic change both have independent effects in the adjustment of root length (i.e., root depth in this study).

As it is difficult to fully measure root architecture, previous studies commonly used topology to describe the altered branching patterns indicative of a facet of the root architectural change (*Fitter, 1987*; *Harper, Jones & Sackville, 1991*; *Lynch, 1995*). Here, we not only used the topological index (*Fitter, 1987*; *Glimskär, 2000*) and the dichotomous branching index (*Šmilauerová & Šmilauer, 2002*), both commonly applied to characterize root topology (i.e., TI and DBI, respectively; see Table 1 for definitions), but also used the TRMF (see Table 1 for definition) to characterize root architectural change.

It is logical to characterize root architecture as the biomass proportion of a certain functional root module in relation to the whole root system (e.g., TRMF) because the root branching pattern just refers to the coordinated growing relation among different functional root modules, and this relation could be represented with biomass proportion as a mass proxy. In particular, an extreme herringbone-like branching pattern has been found to be primarily confined to the main axis (*Fitter et al., 1991*)—that is, possessing the largest proportion of taproot biomass. Therefore, considering taproot as a key root module
functioning in deep rooting, in this study TRMF was specially used to trace the facet of root architectural change that contribute to deep rooting. In addition, the correlation between TRMF and the commonly used TI and DBI has been examined to determine the availability of TRMF to represent root architecture ($R^2$ = 0.643 and 0.698, respectively, both $P$ < 0.001; Fig. S1), and the results indicated that TRMF could be feasibly used in this study.

*Populus euphratica* has an obvious taproot that determines root system depth, and thus absolute root depth can be reflected by taproot length. Accordingly, RRD (see Table 1) was calculated as the absolute taproot length to taproot biomass ratio. Finally, RRD can be factored into RMF, TRMF, and STRL, as follows:

$$RRD = RMF \times TRMF \times STRL \qquad (1)$$

## Data analyses

Differences observed in biomass allocation as well as in root architecture and morphology among the three drought treatments were tested using one-way analysis of variance (ANOVA, Welch's $F$-test). After that, variations of statistical significance were further subjected to post hoc pairwise analysis by applying $t$-tests with Bonferroni corrections, or Games-Howell tests if the homogeneity of variances was not assumed, considering $P$ < 0.05 as significant. The dependence between TRMF and TI or TRMF and DBI was determined by Pearson's correlation analysis. Statistical analyses were performed in SPSS (version 19, SPSS Inc., Chicago, IL, USA).

We calculated the relative contributions of the variance in RMF, TRMF, and STRL to RRD, referencing the variance partitioning method of *Rees et al. (2010)* and *Freschet, Swart & Cornelissen (2015)*. Given that RRD = RMF × TRMF × STRL, our calculation can be expressed as:

$$rrd = rmf + trmf + strl \qquad (2)$$

where the lowercase acronyms indicate $\log_e$-transformed variables (e.g., rrd = ln(RRD)). Thus, the variance decomposition of rrd, for instance, can be expressed as follows:

$$\begin{aligned}
Var(rrd) =& Var(rmf) + Cov(rmf, trmf) + Cov(rmf, strl) + Var(trmf) \\
& + Cov(rmf, trmf) + Cov(trmf, strl) + Var(strl) + Cov(rmf, strl) \\
& + Cov(trmf, strl) \qquad (3)
\end{aligned}$$

Following Eq. (3), as a sample, the contribution of variation in trmf to the variation in rrd can be written as:

$$Cont(trmf) = [Var(trmf) + Cov(rmf, trmf) + Cov(trmf, strl)]/Var(rrd) \qquad (4)$$

Where Var is the variance and Cov is the covariance. This variance partitioning was only performed when substantial variation (i.e., ≥15%) in rrd was observed across treatments, so as to avoid meaningless results.

**Table 2 Mean values of biomass variables, root branching variables and root morphological variables of *P. euphratica* seedlings at 110 days under drought treatments.**

| Variables | | OW (control) | MD | SD |
|---|---|---|---|---|
| Biomass | Total plant biomass (g) | 0.46 ± 0.02[a] | 0.35 ± 0.02[b] | 0.22 ± 0.02[c] |
| | Above-ground biomass (g) | 0.30 ± 0.02[a] | 0.23 ± 0.02[b] | 0.13 ± 0.01[c] |
| | Below-ground biomass (g) | 0.16 ± 0.01[a] | 0.12 ± 0.01[b] | 0.09 ± 0.01[c] |
| | Taproot biomass (g) | 0.043 ± 0.003[a] | 0.032 ± 0.002[b] | 0.030 ± 0.002[b] |
| Morphology | Total root length (cm) | 662 ± 37[a] | 516 ± 44[b] | 393 ± 41[c] |
| | Average root length (cm) | 0.376 ± 0.048[a] | 0.367 ± 0.038[a] | 0.236 ± 0.027[a] |
| | Root diameter (mm) | 0.348 ± 0.007[a] | 0.357 ± 0.006[a] | 0.359 ± 0.009[a] |
| | SRL (cm/g) | 4,766 ± 290[a] | 4,846 ± 357[a] | 5,141 ± 343[a] |
| | Average distal root length (cm) | 0.319 ± 0.122[a] | 0.279 ± 0.191[b] | 0.156 ± 0.057[c] |
| | Distal root diameter (mm) | 0.278 ± 0.006[c] | 0.302 ± 0.007[b] | 0.337 ± 0.007[a] |
| | SDRL(cm/g) | 10,032 ± 425[a] | 9,487 ± 661[ab] | 7,843 ± 361[b] |
| | Average lateral root length (cm) | 5.113 ± 0.324[a] | 4.680 ± 0.333[ab] | 3.765 ± 0.298[c] |
| | Lateral root diameter (mm) | 0.881 ± 0.035[a] | 0.757 ± 0.034[b] | 0.653 ± 0.029[c] |
| | Taproot length (cm) | 35.3 ± 1.1[a] | 27.3 ± 0.8[b] | 26.2 ± 0.7[b] |
| | Taproot diameter (mm) | 1.60 ± 0.091[a] | 1.48 ± 0.093[a] | 1.35 ± 0.082[a] |
| Architecture | TI | 0.713 ± 0.007[b] | 0.731 ± 0.008[ab] | 0.757 ± 0.010[a] |
| | DBI | 0.101 ± 0.010[b] | 0.153 ± 0.016[ab] | 0.216 ± 0.020[a] |
| | Lateral root branching density (*n*/cm) | 1.21 ± 0.09[a] | 1.03 ± 0.05[b] | 0.85 ± 0.04[c] |

**Notes:**

Differences in variables among treatments were tested using factorial analysis of variance (ANOVA). Values are means ± standard error (OW: $n = 25$; MD: $n = 28$; SD: $n = 28$). In each row, means followed by different letters are significantly different ($P < 0.05$). OW, MD, and SD refers to optimal water content (control), moderate drought stress and severe drought stress separately.

SRL, specific root length; SRDL, specific distal root length; TI, topology index; DBI, dichotomous branching index.

## RESULTS

### Drought-induced phenotypic changes in biomass, root morphology, and root architecture

Drought stress inhibited *P. euphratica* seedlings' growth in both biomass and root morphology (Table 2). Drought conditions caused a dramatic decrease in the biomass of different plant parts, total root length, the average length of lateral roots and distal roots, and taproot length. However, when examining responses to differing drought intensities, taproot length, and taproot biomass did not display a consistent declining trend with increasing drought intensity, and there was no difference in taproot length and taproot biomass between MD and SD treatments (Table 2). Moreover, the diameter of different root classifications had different responses to drought (Table 2). Root diameter (RD) (calculated with total roots) did not change significantly under drought conditions, but distal RD increased and lateral RD decreased significantly (Table 2). Taproot diameter decreased with drought, but not significantly. Furthermore, while the SRL (calculated with total roots) did not change by drought, the SRL of the distal roots (i.e., SDRL)—the root components more actively involved in water uptake—showed a significant decrease in SD in relation to OW (Table 2).

**Table 3 Representative indices of different root phenotypic changes and deep-rooting capacity of *P. euphratica* seedlings under drought treatments.**

| Indices | OW (control) | MD | SD |
|---|---|---|---|
| RMF | $0.338 \pm 0.009^b$ | $0.353 \pm 0.013^b$ | $0.418 \pm 0.013^a$ |
| TRMF | $0.304 \pm 0.018^b$ | $0.333 \pm 0.026^{ab}$ | $0.393 \pm 0.030^a$ |
| STRL (cm/g) | $889 \pm 37^a$ | $919 \pm 45^a$ | $1,020 \pm 67^a$ |
| RRD (cm/g) | $90.8 \pm 7.26^b$ | $108.0 \pm 11.0^{ab}$ | $166.0 \pm 15.9^a$ |

Notes:
Differences in indices among treatments were tested using factorial analysis of variance (ANOVA). Values are means ± standard error (OW: $n$ = 25; MD: $n$ = 28; SD: $n$ = 28). Within a row, means followed by different letters are significantly different ($P < 0.05$). OW, MD, and SD refers to optimal water content (control), moderate drought stress and severe drought stress, respectively.
RMF, root mass fraction; TRMF, taproot mass fraction; STRL, specific taproot length; RRD, relative root depth.

*Populus euphratica* seedlings showed a high plasticity in root branching patterns under different drought stresses. DBI and TI measures the degree to which a root is perfectly herringbone (DBI or TI equal to one) or dichotomous branching (DBI equal to zero, or TI close to 0.5). Values of DBI and TI differed significantly among the three drought treatments, but both of them in MD didn't differ from in OW or in SD (Table 2). The ranges of value presented by the total samples were 0.650 to 0.844 for TI and 0.032 to 0.392 for DBI. Lateral root branching density, a simple but direct trait reflecting root branching, showed a markedly decreased trend from OW to SD treatments (Table 2).

## Different levels of root phenotypic variables related to deep rooting changed differently under drought

Drought had a significant positive effect on RRD, but this increase was not proportional to the differences in RRD between SD and MD, it was almost three times greater than the difference in RRD between MD and OW (Table 3). Regarding the indices representing different levels of root phenotypic changes, RMF and TRMF increased markedly with increasing drought severity, while STRL did not significantly increase under drought conditions. In addition, the TRMF under SD was not significantly different from that under MD, but RMF markedly increased under SD in relation to that under MD (Table 3).

## Relative contributions of different root phenotypic changes to deep rooting

These results indicated that drought stress had a positive effect on RRD mainly via significant positive effects on TRMF and RMF. Generally, TRMF and RMF contributed to over 75% of the variation in RRD, while STRL contributed to approximately only 20% (Fig. 3). Moreover, TRMF always had the largest relative contribution to the variation of RRD both under MD (55%) and under SD (41%) in relation to that under OW (Fig. 3). Furthermore, the relative contributions of RMF, TRMF, and STRL to RRD variation changed with drought intensity (Fig. 3). TRMF contributed 55% of RRD variation between MD and OW, which was larger than the 27% contributed by RMF and 18% by STRL. However, under SD, RMF contributed to 37% of the variation in RRD, nearly equal with 41% for TRMF. The contribution of STRL on RRD variation also increased slightly to 22%.

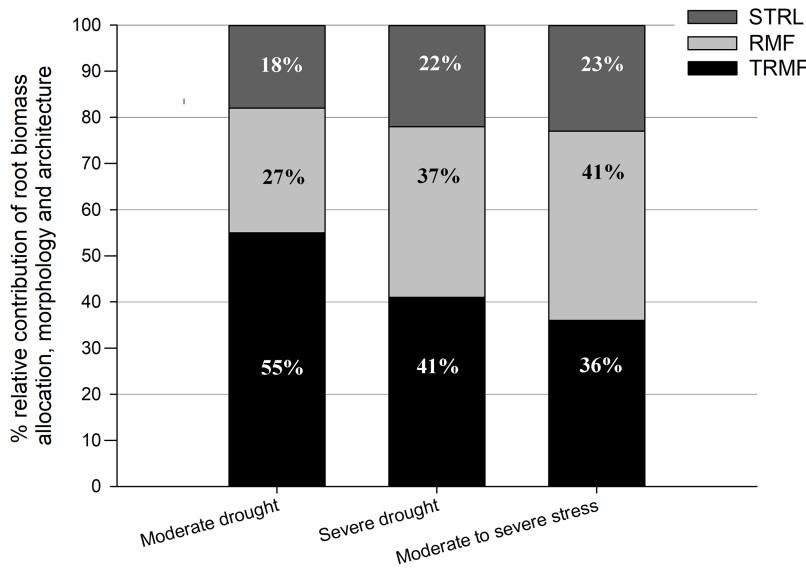

**Figure 3** Relative contributions of root-shoot allocation (RMF, light gray bars), morphology (STRL, dark gray bars) and architecture (TRMF, black bars) variables to the variation in relative root depth (RRD). From left to right, the first bar represents total contributions to the variation in RRD between OW and MD, the second bar represents that between OW and SD, and the third bar represents that between MD and SD. OW, optimal water content (control); MD, moderate drought stress; SD, severe drought stress.

In addition, the variation of RRD from MD to SD treatments was even more attributable to RMF changes (41%) than to TRMF changes (36%) (Fig. 3).

## DISCUSSION

### Root adaptive responses to drought stress in terms of allocation, morphology, and architecture

One of the basic ways for plants to adapt to a shortage of below-ground resources is to maximize fine root area (e.g., decreased RD and increased SRL under drought stress) (*Fitter, 1985*; *Cortina et al., 2008*; *Olmo, Lopez-Iglesias & Villar, 2014*). However, in the present study, the RD and SRL of whole roots exhibited no significant change, but the RD of distal roots increased and SDRL decreased under drought conditions (Table 2). Commonly, roots with a smaller RD and higher SRL is advantageous under drought stress, as the smaller RD conveys higher resistance to root embolism (*Alameda & Villar, 2012*) through the smaller xylem vessel diameter (*Fitter, 1987*), and a higher SRL probably is meant to commit limited carbohydrate supply for extensional growth (*Trubat, Cortina & Vilagrosa, 2006*). The unusual results (increased RD of distal root and decreased SDRL under drought) in this study were likely due to the smaller SRL, which is positively correlated with root life span and respiration rate (*McCormack et al., 2012*), allowing *P. euphratica* seedlings to save considerable energy in dry soil through a low root turnover rate in distal roots. Moreover, thicker fine roots would be able to penetrate into the more compacted soils imposed by soil drying (*Bengough et al., 2005*). In addition, the thicker distal roots of *P. euphratica* seedlings show higher drought resistance to preserve

their vitality in the topsoil of arid-region floodplains where soil water availability is widely fluctuating with the regime of flood and dry seasons (*Leon et al., 2011*).

Plants mostly elongate root internode lengths to adapt to drought environments (*Nicotra, Babicka & Westoby, 2002*) as this also extends the volume of root exploring (*Fitter & Stickland, 1992*). However, in our results, average root length did not significantly change and lateral root length clearly decreased under drought stress (Table 2). Ecologically, floodplain soil possesses a high infiltration rate, which is due to sandy soil layer (several meters in depth) deposited under the 10–50 cm deep surface clay soil layer formed by floodwater sedimentation (*Thevs et al., 2008*). Thus, rooting downward into deep soil is more critical for drought-stressed *P. euphratica* seedlings than rooting in other directions and shortening most of their lateral roots can avoid inefficient investments in the horizontal exploration of soil zone (*Padilla & Pugnaire, 2007*; *Bauerle et al., 2008*).

Root architecture plays a major role in determining root resource-uptake efficiency (*Fitter, 1987*; *Lynch, 2005*). This multidimensional root feature is generally described by measuring root topology (*Harper, Jones & Sackville, 1991*) or branching density/intensity (*Kong et al., 2014*). In the present study, the lateral root branching density was reduced under drought conditions (Table 2). From a functional point of view, the sparse lateral root branching of *P. euphratica* seedlings seems to conserve the high metabolic cost of root construction and maintenance, which can commonly exceed 50% of daily photosynthesis (*Lambers, Atkin & Millenaar, 2002*). Besides, the sparse lateral root branching may reduce the competition for water among the roots of an individual plant (*Fitter et al., 1991*; *Taub & Goldberg, 1996*), which effectively increases the uptake efficiency per unit of lateral root length (*Postma, Dathe & Lynch, 2014*).

Our results indicated that *P. euphratica* seedlings tended to create a herringbone-like branching pattern under drought conditions (Fig. 2), as revealed by the increased TI and DBI values under increased drought treatments (Table 2). This finding is in line with most model-based and empirical studies conducted on plants under drought conditions (*Fitter, 1986*; *Taub & Goldberg, 1996*). Herringbone-like root systems possess higher exploration efficiency (*Ho et al., 2005*; *Paula & Pausas, 2011*), thereby allowing for *P. euphratica* seedlings to reach water-rich deep soils quickly. Additionally, it should be noticed that in our experiment there is a vertical soil moisture gradient in the plant pots. Other studies performed with more actual ground water conditions and different soil texture layers would be an alternative strategy to determine and confirm deep-rooting of *P. euphratica* seedlings.

## Changes in absolute and relative root depth under drought stress

The four-month-old *P. euphratica* seedlings grown in our experiment ultimately presented total biomasses ranging from about 0.2–0.5 g (Table 2), and, surprisingly, they developed taproots with approximately 26 to 35 cm long (Table 2). This indicates that, despite their low biomass accumulation rate, *P. euphratica* seedlings have a great capability to root deeply at their early stages. The miniscule biomass accumulation by the first-year seedlings might be due to their extremely small and light seeds (0.1–0.2 g per 1,000 seeds).

A similar ontogeny was also found in other studies concerning first-year riparian seedlings. An experiment under drought conditions conducted by *Wang et al. (2015)* showed that *P. euphratica* seedlings sown in April had about one g dry mass and over 22 cm taproot length by the end of July. Likewise, a study of riparian tree seedlings (family Salicaceae) conducted by *Stella et al. (2010)* demonstrated that three-month-old cottonwood seedlings (*P. fremontii*) had 0.3 g dry weight with roots over 20 cm in depth, and that three-month-old *Salix exigua* and *S. gooddingii* seedlings developed root depths exceeding 25 and 40 cm, respectively, despite exhibiting dry weights of only 0.22 and 0.4 g, respectively. Clearly, the tiny but deep-rooting seedling phenotype seems common in riparian tree species growing in arid regions.

Changes in plant root depth under drought stress are controlled by two processes (Fig. 1) (*Sultan, 2000*). On the one hand, drought stress weakens photosynthesis, dwindles the accumulated biomass and body size, and thereby leads to a shortening of the root depth. On the other hand, the root depth is influenced by different drought-induced root phenotypic responses. In this study, the absolute root depth of *P. euphratica* seedlings decreased significantly (Table 2) while the RRD increased significantly under drought conditions (Table 3), indicating that the limitation in growth caused by drought had an overwhelming effect on root depth, but which was compensated by enhancing deep-rooting capacity of *P. euphratica* seedlings under drought.

## Relative contributions of root phenotypic changes to the increase of relative root depth

Our results indicated that changes to root architecture and root-shoot allocation in *P. euphratica* seedlings dominated for achieving deep rooting under drought conditions, while the role of morphological changes was minor (Fig. 3). This supports the perspective of *Freschet, Swart & Cornelissen (2015)* that root-shoot allocation was more important than root morphological changes for plant adaptations to changing environmental conditions. Besides, STRL in this study did not significantly increase under drought stress (table 3) and maintained relatively slight contributions to deep rooting regardless of drought intensity (Fig. 3), although taproot stretching seems to be an efficient way to increase root depth. This implies that phenotypic changes in module-level (morphology) seems not much important for plants to respond to belowground resources, given that the role of SRL of fine roots (absorptive roots) are also marginal for plant phenotypic adaption to nutrient limitation (*Kramer-walter & Laughlin, 2017*). The potential negative effects of increased STRL on taproot function may account for this result. Because SRL is determined negatively by the RD and root tissue mass density (*Nicotra, Babicka & Westoby, 2002*), increased STRL means decreased taproot diameter or tissue density. This negatively affects taproot conduction, anchorage, and penetration, which are all essential functions for seedlings that are suffering with drought conditions.

The relative contributions of changes in root-shoot allocation and root architecture were altered under different drought intensities (Fig. 3). Under MD, root architectural changes played a decisive role (contribution over 50%) for increasing RRD. However, under SD, the root-shoot allocation response was stronger than that under

MD (Table 3), and its relative contribution to deep rooting became nearly as important as that of root architectural changes (Fig. 3). This is likely because increased root-shoot allocation would have decreased photosynthetic capacity and accumulation of photosynthates (*Muller et al., 2011*). Root architecture changes seems more carbon economical under drought conditions, by which plant would root deeply only at the cost of weakening exploration capacity to the horizontal and surface soil (*Thevs et al., 2008*). Moreover, the variation of RRD from SD to MD is about three times that of the variation from OW to MD (Table 3), which indicates that keeping deep root is more important when drought becomes severe. Therefore, adjustment in root architecture seems to be insufficient for *P. euphratica* seedlings to root deeply under severe drought conditions. From a functional view, the seedlings possibly cannot easily acquire the adequate amount of water to maintain metabolic processes with increased drought stress. Accordingly, the seedlings would rely more on allocating biomass to roots, which have an advantage in minimizing water loss of shoot transpiration and in enhancing deep rooting potential (*Brunner et al., 2015*). In addition, our study revealed a practical implication for breeding *P. euphratica* seedlings. To promote the survival of container seedlings transplanted in the field, it is a good idea to breed the seedlings under MD before transplantation, in order to promote deeper and steeper root systems while minimally affecting their sizes.

It has been widely reported for many plants, including *P. euphratica* seedlings (*Bogeat-Triboulot et al., 2007*), that root-shoot allocation responds significantly only to severe environmental stresses (*Poorter et al., 2012*). A general explanation is that plants maintain their above-ground growth for as long as possible under moderate soil environmental stresses to keep aboveground competitiveness (*Padilla et al., 2009*; *Poorter et al., 2012*), but how plants tackle moderate belowground resources stress to maintain shoot growth is still unclear. Our results indicate that *P. euphratica* seedlings are able to adapt to MD mainly via phenotypic adjustments within their root systems. Here, we could raise an assumption that before root-shoot allocation strongly responds to severe drought (*Poorter et al., 2012*), change in root architecture mainly shoulder the absent role of root-shoot allocation for drought-suffered plant. This hypothesis is based on a premise that plant phenotypic changes at the organ-level are more carbon economical than that at the individual-level, while the latter should be more water economical than the former for drought adaptation. To confirm this hypothesis, more functionally-different species need to be further studied.

## CONCLUSIONS

(1) *Populus euphratica* seedlings showed a conservative resource-use strategy in response to drought stress, evidenced by thicker and shorter distal roots with lower SRL, sparser lateral root branching, and herringbone-like root architecture under drought.

(2) The absolute root depth of *P. euphratica* seedlings was strongly constrained by drought, but this negative effect was alleviated by changes in root-shoot allocation, root architecture, and taproot morphology, resulting in a significant increase in RRD.

(3) Root architectural changes and root-shoot allocation dominated in order to achieve deep rooting under drought conditions, while the role of taproot morphological changes was relatively minor. Interestingly, their relative contributions to deep rooting varied with drought intensity. Under moderate drought conditions, root architectural changes exerted a predominant effect on increased RRD, but under severe drought, root-shoot allocation and root architecture played equally important roles.

## ACKNOWLEDGEMENTS

We are profoundly thankful to Ejin Banner State Forest Farm for their support for conducting this experiment, and to our colleagues Liu W., Dong. F.Y. and Zhong Y.M. for their assistance in the field and indoors. We also thank the public infrastructure laboratory in the College of Forestry of Beijing Forestry University for providing laboratory instruments for use.

### Funding

This work was supported by the National Natural Science Foundation of China and State Forestry Administration research special funds for public welfare projects (Grant No.: 31570610 and 201404304, respectively). The funders had no role in study design, data collection and analysis, decision to publish, or preparation of the manuscript.

### Grant Disclosures

The following grant information was disclosed by the authors:
National Natural Science Foundation of China.
State Forestry Administration research special funds for public welfare projects: 31570610 and 201404304.

### Competing Interests

The authors declare that they have no competing interests.

### Author Contributions

- Zi-qi Ye conceived and designed the experiments, performed the experiments, analyzed the data, prepared figures and/or tables, authored or reviewed drafts of the paper, approved the final draft.
- Jian-ming Wang performed the experiments, analyzed the data, prepared figures and/or tables, authored or reviewed drafts of the paper.
- Wen-juan Wang performed the experiments, contributed reagents/materials/analysis tools.
- Tian-han Zhang performed the experiments, contributed reagents/materials/analysis tools.
- Jing-wen Li conceived and designed the experiments, contributed reagents/materials/analysis tools, authored or reviewed drafts of the paper.

## Data Availability

The raw data are available in a Supplemental File.

## Supplemental Information

Supplemental information for this article can be found online at http://dx.doi.org/10.7717/peerj.6513#supplemental-information.

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
