# Peer review of "Effects of root phenotypic changes on the deep rooting of Populus euphratica seedlings under drought stresses"

_PeerJ, doi:10.7717/peerj.6513_

## Round 0.1 · original submission · Major Revisions

Reviews suggest that major revisions are needed before the manuscript can again be considered for publication in PeerJ.

Overall, a complete revision is needed with carefull attention to clear English usage. Additionally, clarification of the objectives, results and conclusions of the statistical elements of the manuscript is needed.

Reviewer #3 has laid out the revisions with suggestions and has attached an annotated document for your convenience.

Please address each point raised by the reviewers and clearly document your revisions in an attached track-change document.

We look forward to receiving your revised manuscript.

Reviewer 1 ·

Basic reporting

The oveall structure, the way of presenting, the reasoning, and the English of this MS is well done.

Experimental design

pot-grown seedlings was the basis of this experimental set-up. While there is nothing wrong in doing-so, it should be noted that the results and conclusions are limited within this set-up, and might be rather different from real field conditions.

Validity of the findings

validated within their own context.

Additional comments

the limitation of the experimental set-up should be note in discussion and conclusion.

·

Basic reporting

No comments

Experimental design

No Comment

Validity of the findings

No Comment

Additional comments

One Correction:
1) In Line 297, it is reported as Wang (2015), but there is no reference for this in the reference section. I guess, this should be Wang et al., 2015.


Suggestion:
1) The supplemental figure with the scanned images under different drought conditions is pretty nice one. It could be included in the text as supplemental fig; Not sure if the authors forgot to include about this in their text.

Reviewer 3 ·

Basic reporting

The study was a pot experiment, which supplied the adaptive regualtion information of root morphology,distribution and architecture under differnter drought treatments. On the whole, the manuscript did not present novel findings in plant grwoth adaption area. There are some mistakes in figs and tables presentation, and the english also need to improve by a native english speaker.

Experimental design

The experiment want to distingush the relative contribution of root morphology, architectue and biomass allocation on root adjustation to drought stress, but it is hardly to address this, due to their autocorrelation. In addition, if the author want to determine the improtance of deep root to suction of underground water, a serial of underground water level shoud be include in the experiment.The define of the relative tap root length is not appreciate.

Validity of the findings

the manuscript distingusished source of variation of deep root under drought, whihc is usful for evaluating plant morphological regulation under unfavoriable environmental change.

Additional comments

This is a simple but valuable study, the author tried to explore the relative contributions of root morphology, architecture and carbon allocation in root adaptation to drougt. Buts this is not a well-written paper, evidencing by unclear organization and figs and tables presentation.

Annotated reviews are not available for download in order to protect the identity of reviewers who chose to remain anonymous.

Reviewer 4 ·

Basic reporting

Some suggestions were made in this section aiming to improve the manuscript and are described individually in the general comments.

Experimental design

No comment

Validity of the findings

No comment

Additional comments

This paper presents a study using Populus euphratica seedlings to compare root phenotypic changes in two drought conditions in relation to a control. This study seems to be relevant for the understanding of mechanisms undertaken by this species to overcome drought effects in the plant. The manuscript is well structured and the results seem consistent. There are some considerations that could help to improve and let the manuscript more clear to the readers. Also, the interpretation of some statistical analysis need to be done more carefully, as suggested bellow.

I recomend to consider the following issues:
1. A few sentences could be re-written to make more sense or to improve the overall article writing. I tried to re-write some of them, which can be found in the PDF file as comments or marked in red or blue (associated with comment notes). Also, some suggestions were added to the text, indicating where other minor changes should be made.
2. The legend of figures and tables need to be sufficient for the readers to comprehend the data. So, it is necessary to define the abbreviations and the statistical analysis used in each figure/table and make this very clear.
3. The authors need to cite the figures and tables when they describe the results and in the discussion, otherwise, the reader won’t follow the ideas.
4. Figure 1 needs improvements to be kept in the article. For example, the aerial part of the plant shouldn't be the drawing of a heart. In the end of each sentence there is an arrow. Besides that, the arrow from the middle is covering the text. The use of a proper image editor software might be a very useful tool to construct the image again.
5. The tables 2, 3 and 4 could be combined in one single table, since they show variables in the same treatments and with the same statistical analysis.
6. For a better organization, the root variables in Table 1 should be described in the order of appearance in the text. Besides that, the overall organization of the table needs improvement. For example, the units shouldn't be in separate lines; sentences shouldn't begin with lower case. The table needs to be reviewed carefully.
7. Column 1 of the table with the raw data (supplemental material) needs an adequate identification. It is written “numbers”, but do these numbers refer to the seedlings? The authors must let this clear. In column M is it SDRL or SRL? A review of the columns titles is necessary to make a good presentation of the data.
8. The figure of the supplemental material is very illustrative and helps to comprehend the manuscript. However, it isn’t cited in the text. I think that this figure could be included in the body of the article, not as a supplemental material. A scale needs to be included in the figure and some modifications could be done in the figure legend, as follows: “Representative scan images used in the study to illustrate changes in root system of P. euphratica seedlings with drought treatments”.
9. The data is statistically controlled. However, the authors need to take more care when analyzing the results and interpreting the statistical significance. Data sharing the same letter aren’t statistically different and the interpretations must take this into account. For example, the lines 203-205, 221-223 and 227 should be re-written according to the statistical significance of the data (see notes in PDF file). In addition, when there isn’t statistical difference this should be indicated with “ns” in the data of the tables.
10. In the topic “Experimental treatments” of Materials and Methods, the authors need to take a look at the temperature values indicated, because they don’t make sense. About the replications, each experiment was replicated 30 times or 30 seedlings were evaluated per treatment? Also, it is specified that 85 seedlings were used in the total, but there are data for only 81 seedling in the table with the raw data. These numbers need to be corrected also in the legends of the tables, where the authors indicate the “n” of each treatment.
11. The sentence of lines 153 and 154 is a bit messy and hard to understand. I believe that the authors could write it better.
12. In the lines 278-280, the authors should state that there is a tendency of change to herringbone, but the data (especially DBI) indicated a dichotomous-like branching in the moment of the evaluations.
13. Lines 289-290: Is it a title with subtitle? Or is it the same sentence? Please make it clear.
14. In the line 362 it would be better if the authors could state to what kind of economy they are referring to.

Annotated reviews are not available for download in order to protect the identity of reviewers who chose to remain anonymous.

---

## Round 0.2 · Minor Revisions

Thank you for revising the manuscript according to reviewer comments. Reviewer #4 has made additional minor suggestions for revision; please be careful to incorporate them in your final revision. Your manuscript will be ready for acceptance, once we receive the revised and tracked changes document.

·

Basic reporting

- The authors have carefully addressed the issues and have revised the manuscript very well.
- Good revision on the tables, and the legends.
- The work represented by the authors can lead to root-phenotype studies in other plant species.
-

Experimental design

- The authors have revised the experimental design more organized way.
- Easy to understand the experimental design.

Validity of the findings

- The data is good.
- Good representation of the data.

Additional comments

Well representation of the data, and very well revised. The authors have addressed the issues raised previously by the reviewers, and a great job on revising the manuscript.

Reviewer 3 ·

Basic reporting

The revised paper has been improved.

Experimental design

The author provided more details of the experiment in the revised paper. It is more clear than the previous one.

Validity of the findings

It would be better if the same data analyses been done on the field investigation.

Additional comments

The author has made a careful revision of the last review opinion, which makes the current text more readable. Because there are few studies on the root architecture of natural plants, this paper may provide some ideas for researchers.

Reviewer 4 ·

Basic reporting

The authors have performed the modifications suggested in the first review of the manuscript and it was substantially improved.

Experimental design

No comment

Validity of the findings

No comment

Additional comments

I have only some minor modifications to suggest:

1. Line 39: Correct the word “acquiring”.
2. Line 58: What do the authors mean with "except"?
3. Line 163: I suggest to change this sentence to: "Root phenotypic adjustments and deep-rooting capacity analysis".
4. Line 168: There is no need to cite Fig. 1 here.
5. Line 185: Correct the word “representing” to “represent”.
6. Lines 206 and 213: Correct the position of the “comma”.
7. Line 220: Correct “(Table. 2)” to “(Table 2)”.
8. Line 221: Where is the average length of taproot shown in the table?
9. Line 229: Correct the word “changed” to “change”.
10. Line 235: MD didn't differ from OW or SD.
11. Line 254: Correct to: “contributed to approximately only 20%”.
12. Line 305: It is a good moment to cite Fig. 2 (after “drought conditions”).
13. Line 310: I suggest to consider the following way to write this sentence: “Additionally, it should be noticed that in our experiment there is a vertical soil moisture gradient in the plant pots. Other studies performed with more actual ground water conditions and different soil texture layers would be an alternative strategy to determine and confirm deep-rooting of P. euphratica seedlings.”
14. Line 315: Changes to in absolute
15. Line 333: “thereby leads to a shortening of the root depth”
16. Line 349: “plants to respond to belowground resources”
17. Line 355: “functions for seedlings that are suffering with drought conditions.”
18. Line 357: Remove the word “obviously”.
19. Line 360: Maybe this sentence could be split in two: “…photosynthates (Muller et al., 2011). Root architecture changes…”
20. Line 406: Correct the word “colleague” to “colleagues”.
21. Fig. 1: remove the dots between the words (inside the boxes).
22. Fig. 2: Correct in the title: (C) & (D) from the seedlings under MD
23. Fig. 3: Ideally, the legend with the colors should be shown according to the order of appearance in the graph: STRL (dark gray) on the top and TRMF (black) on the base.
24. Table 2: Title: Correct to: “Mean values of biomass, root morphological and root branching variables of…”

---

## Round 0.3 · accepted · Accept

The detailed revisions noted by reviewer #4 have been carefully implemented. Thank you. Your manuscript is now accepted for publication in PeerJ.